# The Maternal and Fetal Outcomes of Pregnancy in Wilson’s Disease: A Systematic Literature Review and Meta-Analysis

**DOI:** 10.3390/biomedicines10092072

**Published:** 2022-08-24

**Authors:** Tomasz Litwin, Jan Bembenek, Agnieszka Antos, Iwona Kurkowska-Jastrzębska, Adam Przybyłkowski, Marta Skowrońska, Łukasz Smoliński, Anna Członkowska

**Affiliations:** 1Second Department of Neurology, Institute of Psychiatry and Neurology, Sobieskiego 9, 02-957 Warsaw, Poland; 2Department of Clinical Neurophysiology, Institute of Psychiatry and Neurology, Sobieskiego 9, 02-957 Warsaw, Poland; 3Department of Gastroenterology and Internal Medicine, Medical University of Warsaw, Banacha 1a, 02-097 Warsaw, Poland

**Keywords:** Wilson’s disease, fertility, copper, pregnancy, neurological symptoms, birth defects, systematic review

## Abstract

Wilson’s disease (WD) is a rare, treatable genetic disorder with multi-organ symptoms related mainly to copper accumulation. Most patients become aware of the disease as young adults, thus knowledge on fertility, pregnancy course and outcome is very important both for patients and physicians. The aim of this study was to perform a systematic review and meta-analysis of pregnancy outcomes in women with WD. This systematic literature review was performed according to Preferred Reporting Items for Systematic Reviews and Meta-Analyses (PRISMA) guidelines. Studies were identified by searching the PubMed database (up to 12 January 2022) and by screening reference lists. We found 49 publications, including 13 retrospective studies and 36 series and case reports on pregnancy outcomes in WD patients. In total, descriptions of 449 pregnant women with 822 pregnancies were retrieved. Successful deliveries were achieved in 78.3% (644/822) of all pregnancies. Spontaneous abortions were observed in 21.7% (178/822) of pregnancies, more frequently in patients who were untreated 68.6% (96/140). Analyzing maternal outcome, 2.2% (18/822) of pregnancies were associated with the aggravation of neurological symptoms. Symptoms of hepatic deterioration were observed in 4.6% (38/822) of cases. These were usually transient and recovered after pregnancy; however, death due to liver failure was observed in 0.2% (2/822) of cases. Birth defects occurred in 4.7% (39/822) of pregnancies. The available meta-analysis showed statistically significant positive associations between anti-copper treatment and pregnancy outcome. Our results document the significance of anti-copper treatment as the main factor leading to successful pregnancy, as well as positive outcomes for women with WD.

## 1. Introduction

Wilson’s disease (WD) is an inherited disorder of copper metabolism with copper accumulation in different organs and secondary damage of the affected tissues [1,2,3,4,5]. In concordance with WD pathogenesis, most patients present with hepatic manifestations (from clinically asymptomatic liver enzymes increase, through to hepatitis, liver cirrhosis as well as acute liver failure), with some patients experiencing neurological symptoms (such as a wide spectrum of movement disorders) and additionally, patients may present with mostly mild psychiatric symptoms [1,2,3,4,5]. WD is potentially treatable with anti-copper agents; however, treatment has to be introduced at an early phase of the disease (without severe liver or brain injury) and has to be lifelong, with compliance being essential for treatment outcome [1,2,3,4,5]. As WD is usually diagnosed in young adults, including women with childbearing potential, knowledge about pregnancy outcomes, the use of anti-copper treatment during pregnancy and breastfeeding, as well as potential teratogenic effects of treatment is important for women with WD and physicians [1,2,3,4,5].

Even in healthy women, physiological and hormonal changes during pregnancy may lead to changes resembling alterations that may occur in the course of liver disease, such as increased circulatory volume, cardiac output, maternal heart rate and decreased peripheral vascular resistance—similar to decompensated liver disease [6]^.^ Small esophageal varices may be observed as a result of compression of the vena cava inferior by the growing uterus, which reduces venous return. Increases in estrogen in particular, but also progesterone and other hormones including human chorionic gonadotropin, prolactin, cortisol and human placental lactogen may also be observed, which can additionally affect liver function as well as copper metabolism [1,6]. Moreover, several liver disorders related only to pregnancy may occur, including hyperemesis gravidarum, gestational intrahepatic cholestasis, acute fatty liver of pregnancy and liver diseases related to hypertension (preeclampsia/eclampsia, subcapsular hematoma and liver rupture or HELLP syndrome (hemolytic anemia, elevated liver enzymes, and decreased platelet count)) [6]. Finally, some WD patients present with neurological deficits or psychiatric symptoms, which can impact the pregnancy and the delivery by itself or through their symptomatic treatment [7].

A limited number of retrospective studies and case/series reports [7,8,9,10,11,12,13,14,15,16,17,18,19,20,21,22,23,24,25,26,27,28,29,30,31,32,33,34,35,36,37,38,39,40,41,42,43,44,45,46,47,48,49,50,51,52,53,54,55] have evaluated maternal and newborn outcomes in WD, and here, we aimed to analyze these studies to provide a more comprehensive overview.

## 2. Methods

This systematic review was performed in concordance with the international accepted criteria of the Preferred Reporting Items for Systematic Reviews and Meta-analyses (PRISMA) statement [56].

### Search Strategy, Eligibility Criteria and Meta-Analysis Methodology

We searched the PubMed database (up to 12 January 2022) for original studies (prospective and retrospective), and case and series reports analyzing pregnancies and their outcomes in patients with WD. Search terms included: (“Wilson’s disease/”Wilson disease” and “pregnancy”) and (“Wilson disease”/”Wilson disease” and “birth defect”). Studies eligible for further analysis were: (1) conducted with humans; (2) original studies (prospective or retrospective); (3) case and series reports of pregnant WD patients; (4) those written in the English language. The reference lists of extracted publications were also searched.

Firstly, title and abstracts were screened independently, according to study criteria by all authors (reviewers); duplicate records were removed. Incomplete reports, reviews, editorial, commentaries, conference proceedings, discussions, as well as overlapped data were excluded after extensive assessment and revisions.

Then, full texts were obtained and screened with the same mode.

Duplications, incomplete reports, reviews, editorials, commentaries, conference proceedings, discussions, and overlapped data were excluded.

All the identified studies were analyzed and verified independently by all authors to confirm the inclusion criteria and were grouped as: (1) prospective studies which aimed to analyze the course of pregnancies and their outcomes; (2) retrospective studies mostly presenting data from country registries of rare disease, according to pregnancies and their outcomes; (3) series reports; (4) case reports of pregnant women with WD. Additionally, a random effects meta-analysis was used to pool data from studies comparing the risk of spontaneous abortion between treated and untreated patients with WD (“meta” package for R). Heterogeneity was assessed with I2 and the chi-squared test for the Cochrane’s Q statistic.

The study protocol was registered and received INPALSY registration number: INPALSY202280003 (doi: 10.37766/inpalsy2022.8.00003)

## 3. Results

The study search and selection process are presented in Figure 1.

Initially, we found 3529 records in total from PubMed searches. After duplicate papers removal, 480 publications remained. The title, abstracts and full texts were then screened for relevance, removing another 431 records. In total, 49 full-text articles on 449 patients (822 pregnancies) were included in the analysis.

There were no prospective studies. There were 13 retrospective studies presenting the results of pregnancies in WD patients based mainly on national WD databases (Table 1).

There were also 36 case reports and case series presenting detailed descriptions of pregnant women with WD as well as maternal and pregnancy outcomes (Table 2).

Successful deliveries were achieved in 78.3% (644/822) of all pregnancies. Spontaneous abortions were observed in 21.7% of all pregnancies (178/822), more frequently in patients who were untreated (68.6% [96/140]) compared with those who were treated. Analyzing maternal outcome, there was aggravation of neurological symptoms in 2.2% of pregnancies (18/822) and of hepatic symptoms in 4.6% (38/822). Exacerbation was usually transient and recovered after pregnancy; however, death due to liver failure was reported in 0.2% (2/822) of cases. Birth defects occurred in 4.7% (39/822) of pregnancies.

We were able to analyze four studies that compared the risk of spontaneous abortion between treated and untreated patients with WD [7,8,10,16]. A random effects meta-analysis of the studies showed that the risk of spontaneous abortion was reduced by 52% in treated patients (risk ratio = 0.48; 95% confidence interval, 0.34–0.67) (Figure 2). There was no significant heterogeneity between the studies (I^2^ = 0%, *p* = 0.67).

## 4. Discussion

To our best knowledge, this is the first systematic review of the literature of patients with WD to present pregnancy outcomes as well as birth defects. As observed in the meta-analysis, the very high rate of spontaneous abortions in untreated patients versus treated women emphasizes the significance of anti-copper treatment before and during pregnancy. The rate of spontaneous abortion in treated WD patients was quite similar to that seen in healthy populations (21.7%) [11]. Consistent with the findings from the retrospective studies, case reports also highlight the significance of WD treatment during pregnancy. The study by Oga et al. [22] described a patient who stopped treatment 9 years before pregnancy where there was documented copper accumulation in the placenta (on the maternal side); higher copper levels in the amniotic fluid (seven times that in healthy women); as well as WD-like symptoms in the neonate, namely hepatomegaly with elevated liver enzymes with a high excretion of urinary copper. These data support that it is not only maternal outcome that is affected by incorrect anti-copper treatment during pregnancy, but newborns may also be affected by copper toxicity.

Our data strongly support the recommendation of the European Association for the Study of the Liver (EASL) for WD from 2012, which advocates that treatment for WD should be continued during pregnancy (Grade II-3, B, 1) [1]. These recommendations also advise a dose reduction for DPA and trientine from the first trimester; however, this guidance is mainly based on speculation rather than on data [1]. Despite our analysis of the all evidence to date, due to limited data, we are not able to support this recommendation further.

The aggravation of neurological symptoms of WD occurred in just 2% of patients, which may be due to stopping anti-copper drugs during pregnancy or dose reduction, but in most cases, the exacerbation was reversible after delivery and with the re-introduction of full-dose anti-copper treatment. There were also cases described with neurological symptoms recovery during the pregnancy and adequate WD treatment. Liver symptoms were aggravated in 4.6% of patients, highlighting the need to carefully monitor patients, as recommended in pregnant patients with liver cirrhosis [6].

Fetal defects occurred in 4.7% pregnancies, which is more frequent than in healthy populations [11]. However, it should be noted that we summarized data from 1959 to the present day, and more recent studies showed a lower number of birth defects, since healthcare has improved [1]. In the study by Pfeiffenberger et al. in 2018, [11] birth defects were only observed in up to 3.3% of newborns, which is similar to the general population [6].

Historically, before Walshe introduced DPA in 1956, [1] women with WD more frequently experienced infertility, spontaneous abortions and stillbirth [1,2,3,4,5,6]. As WD treatment was introduced, it improved fertility and pregnancies outcome [1,2,3,4,5,6,57,58,59,60,61,62,63]. Just after the introduction of DPA, Sherwin et al. [42]. published the case of a 24-year-old woman treated with DPA who had a successful pregnancy, highlighting no maternal or fetal adverse outcomes and even diminished neurological symptoms. During pregnancy, changes in copper metabolism such as increased serum levels of ceruloplasmin may have a protective effect on WD aggravation or symptoms in the child [34].

The first contemporary study with substantive numbers (46 women and 107 pregnancies) was performed in 2000 by Tarnacka, et al. [8]. The abortion/labour ratio in WD patients was found to be higher in untreated (1:3.8) and treated patients (1:5.5) than in the general Polish population (1:10). Moreover, the stillbirth ratio was higher in untreated (4.8%) and treated pregnancies (4%) than the Polish general population (0.1%). There was no evidence of increased teratogenicity during pregnancy between treated and untreated patients.

In the study performed by Pfeiffenberger et al. in 2018, [11] which involved the largest group of WD patients and pregnancies (136 women and 282 pregnancies), there was a high rate of spontaneous abortions (40.0%) in untreated WD patients and those who stopped anti-copper treatment during pregnancy compared with the general populations (10–20%). Analyzing different anti-copper drugs, the authors found a lower rate of abortions in all groups, apart from the trientine group (27.7% abortions), with the lowest abortion rate on zinc (10.0%). In a post hoc analysis, only DPA treatment diminished the abortion rate compared with non-treated WD patients. Fetal malformations were observed in 3.3% of babies from mothers on DPA, 2.7% on trientine (comparable with the general population) and none on zinc. Moreover, the authors identified the following risk factors at the time of conception for spontaneous abortion in WD: the presence of neurological symptoms (40% of abortions) and trends according to portal hypertension (29% of abortions) and liver cirrhosis (26% of abortions). Further observational studies in large groups of WD patients are needed to verify the association between birth defects and different types of anti-copper drugs, as well as different anti-copper treatment strategies (decreased dose, drug changes or pauses) [1,2,3,4,5,6,61,62,63].

### Limitations

Our study has some limitations. Firstly, all of the studies and case reports were retrospectively collected over a long time period (almost 60 years), which has seen many changes in medical care. In most papers, pregnancy and maternal outcomes were well described, but data were missing in some cases, such as details on patient history and treatment regimens.

Moreover, in one of the oldest studies presenting the successful outcome of 29 pregnancies in 18 WD patients, in two patients, the data about multiple spontaneous abortions were provided (without details) [14]. However, the cause of the abortions could not be added to the analysis. Additionally, the studied groups were heterogenous, particularly with respect to treatment. Due to limited data, an analysis of fetal and maternal outcomes was not possible among patients who stopped treatment during pregnancy versus those who continued treatment.

## 5. Conclusions

Our study documented the good outcome of pregnancy in WD patients on anti-copper drugs with a very high rate of spontaneous abortions in untreated patients. The risk of aggravation of WD symptoms during pregnancy appeared low; however, due to physiological changes that can exacerbate hepatic injury during the course of pregnancy, liver function should be monitored in patients with WD before and during pregnancy. The number of birth defects in treated patients appears to be relatively small and should not cause treatment discontinuation, given its crucial role for maternal and pregnancy outcome.

## Figures and Tables

**Figure 1 biomedicines-10-02072-f001:**
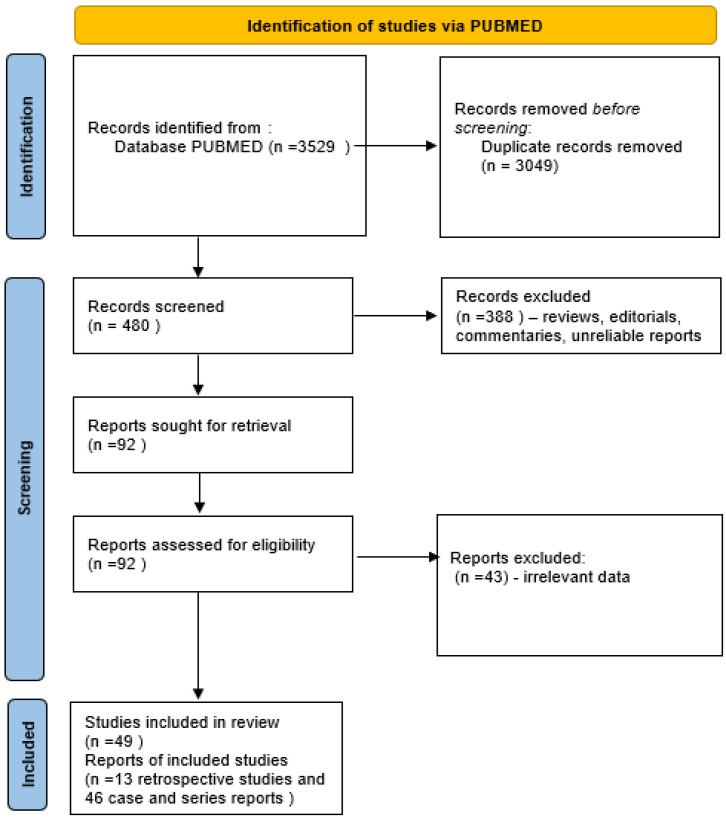
Flow chart of the systematic literature search according to PRISMA guidelines. A total of 3529 articles were found during the initial screen and 49 articles were included in the qualitative synthesis.

**Figure 2 biomedicines-10-02072-f002:**
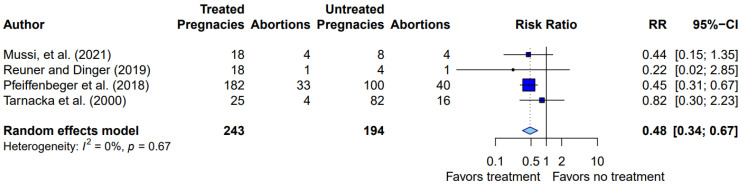
Random effects meta−analysis of studies comparing the risk of spontaneous abortion between treated and untreated patients with WD (squares show risk ratios; square size represents study weight; error bars show 95% confidence intervals; diamond shows pooled effect).

**Table 1 biomedicines-10-02072-t001:** Summary of studies of pregnant patients with Wilson’s disease according to the time of diagnosis, anti-copper treatment, pregnancy and delivery complications, maternal and newborn outcomes, and birth defects.

Reference	Study Details	Patient Characteristics	Outcome: Pregnancy and Maternal	Outcome: Newborn
Mussi et al. (2021) [16]	12 patients and 26 pregnancies Retrospective analysis	Treated pregnancies: 18 DPA: 4 Zinc: 14 Untreated: 8	**Pregnancy outcome**: Successful: 12/26 (46.1%) Spontaneous abortion: 8/26 (30.8%) Untreated 4/8 (50.0%) Zinc 2/14 (14.2%) DPA 2/4 (50.0%) **Deliveries**: Type not reported **Maternal outcome**: Not provided	**Fetal malformations**: 6/18 (33.3%); all on zinc 6/14 (42.8%) **Birth defects**: Corpus callosum agenesis; multiple malformations; left vesicoureteral reflux with hydronephrosis; bicuspid aorta; interatrial communication; several birth defects
Roseira et al. (2020) [17]	18 patients and 19 pregnancies Case-control study (healthy and hepatitis C)	Treated patients: 18 DPA: 14 Zinc: 4 11 hepatic and 7 neurological phenotypes Mean age: 32.7 years	**Pregnancy outcome** : Successful: 12/19 (63.1%) Spontaneous abortion: 7/19 (36.8%) **Deliveries**: Natural **Maternal outcome**: Without deterioration of WD symptoms	**Birth defects**: 3 birth defects (cleft lip and 2 cases of atrial septal defect); 7 cases of low birth weight
Reuner and Dinger (2019) [8]	22 patients and 32 pregnancies Retrospective analysis	Treated prior and during pregnancy: 18 DPA: 14 (30% dose reduction) but 1 discontinued treatment with DPA during pregnancy due to the death of mother and fetus) Trientine: 4 Zinc: 1 Untreated undiagnosed: 3	**Pregnancy outcome**: Spontaneous abortion: 1 patient (3.1%) who decided to stop DPA **Deliveries**: 26 natural; 5 Cesarean sections **Maternal outcome**: Unchanged status: 18 (81%) Hepatic deterioration: 1 (4.6%) Neurological deterioration: 1 (4.6%) Psychiatric deterioration: 1 (4.6%) Death: 1 (4.6%) (patient stopped DPA)	**Birth defects**: None Feeding problem/pure sucking: 9 (28.8%) Ventilatory support: 3 (9.6%) Breast feeding: 27 (86.4%)
Xu-En et al. (2019) [9]	75 patients and 117 pregnancies Retrospective analysis of 75 women with WD and 22 healthy controls	Stopped treatment during pregnancy: 97 No data available according to type of treatment type Mean age: 27.7 years	**Pregnancy outcome**: Successful: 108 (92.3%) Spontaneous abortions: 9 (7.7%) Pregnancy complications: 10 (5 premature membrane ruptures, 2 hydramnios; 3 oligohydramnios) **Deliveries**: 67 natural, 41 Cesarean sections **Maternal outcome**: Deterioration of neurological symptoms: 10/75 (13.3%) Hepatic deterioration: 18/75 (24.0%)	Comparable Apgar scale between WD infants and healthy controls Lower average infant birth with WD vs. controls: 3410 g vs. 3640 g
Pfeiffenberger et al. (2018) [11]	136 patients and 282 pregnancies Retrospective analysis	Treated pregnancies: 182 (64.5%) DPA: 118 (41.8%) Chelator plus zinc: 8 (2.8%) Zinc: 20 (7%) Trientine: 36 (12.7%) Treatment discontinued during pregnancy: 14 (4.9%) No therapy/no diagnosis before pregnancy: 86 (30.4%)	**Pregnancy outcome**: Successful: 209/282 (74.1%) Abortions: 73/282 (25.8%) Abortion rate: Zinc: 10.0% DPA: 16.9% Trientine: 27.7% Paused treatment: 35.7% Undiagnosed WD: 40.6% **Deliveries**: Type not reported **Maternal outcome**: Unchanged status: 263/282 (93.2%) Hepatic deterioration: 16/282 (5.6%) Neurological deterioration: 3/262 (1.1%)	**Births defects**: 7/209 (3.3%) including esophageal atresia; mental retardation and blindness; persistent foramen ovale with thyroid agenesis; 2 atrial septal defects, spastic paralysis; glucose-6-phosphatase dehydrogenase deficiency **Estimated ratio of birth defects**: DPA: 3.3%; trientine: 2.7%; zinc: 0%; 1 in undiagnosed WD and 1 in person who stopped treatment
Dathe e1 al., (2016) [12]	20 patients and 20 pregnancies Retrospective analysis	Treated patients: 20 DPA: 17 (12 patients received 300–1200 mg; 5 patients additionally received zinc) Trientine: 3 (1 patient 1500 mg; 1 patient 300–900 mg; 1 dose unknown)	**Pregnancy outcome**: Successful: 13 on DPA (76.4%) Spontaneous abortions: 3 on DPA (17.6%) Elective termination: 1 on DPA (5.8%) **Deliveries**: Type not reported **Maternal outcome**: Transient thrombocytopenia: 1 Premature membranes rupture with fever: 1	**Birth defects**: 6/13 (46.1%) including 1 umbilical hernia; 1 fetal tachycardia; 1 trisomy 21; 1 scalp defect; 1 head circumference and transient swallowing disturbances; 1 cerebral seizures (transient)
Sinha et al. (2004) [15]	16 patients and 59 pregnancies Retrospective analysis of 341 WD patients, 102 women; 74 women of reproductive age	Untreated: 10 (presymptomatic) Treated with DPA (250 mg/day) and zinc (1320 mg/day): 6 patients with neurological WD (additionally, 2 patients treated with trihexyphenidyl and 1 with phenobarbirate)	**Pregnancy outcome:** Successful: 30/59 (50.8%) Spontaneous abortion: 24/59 (40.6%); 20 in 10 untreated and 4 in treated Medical terminations: 2/59 (3.3%) Still birth: 3/59 (5.0%) **Deliveries**: Type not reported **Maternal outcome**: Severity of WD did not change during pregnancy	**Birth defects**: 30 full-term and delivered healthy
Brewer et al. (2000) [13]	19 patients and 26 pregnancies Retrospective analysis	Neurological WD: 6 (10 pregnancies) Hepatic WD: 13 (16 pregnancies) All patients treated with zinc	**Pregnancy outcome**: Successful: 26/26 (100.0%) **Deliveries**: Type not reported **Maternal outcome**: Good without WD neurological or hepatic deterioration	**Birth defects**: 1 heart defect; 1 microcephaly
Tarnacka et al. (2000) [8]	46 patients and 107 pregnancies Retrospective analysis of 46 women and 27 men with WD (fertility assessment)	Untreated undiagnosed: 31 (82 pregnancies; mean age 22.5 years) Treated: 15 (25 pregnancies; mean age 26.2 years; mean treatment duration of 6 months to 3 years) DPA: 10 (constant treatment in 4 patients; 2 patients halved dose; 4 patients discontinued treatment themselves, with 1 exacerbation of disease) Zinc: 5 (continued therapy)	**Pregnancy outcome**: **Untreated group**: Spontaneous abortions: 8/31 (25.8% patients and 16/82 (19.5%) pregnancies Imminent abortions: 3/31 (9.6%) patients and 2/82 (2.4%) pregnancies Gestosis: 3/31 (9.6%) patients and 6/82 (7.3%) pregnancies Preterm births: 3/31 (9.6%) patients and 3/82 (3.6%) pregnancies Stillbirths: 2/32 (6.4%) patients and 4/82 (4.8%) pregnancies **Deliveries**: 59 spontaneous, 2 Cesarean sections and 1 vacuum extractor **Treated group**: Spontaneous abortions: 4/15 (26%) patients and 4/25 (16%) pregnancies Imminent abortions: 5/15 (33.3%) patients and 5/25 (20%) pregnancies Gestosis: 1/15 (6.6%) patients and 1/25 (4%) pregnancies Preterm births: 3/15 (20%) patients and 3/25 (12%) pregnancies **Deliveries**: Type not reported **Maternal outcome**: Exacerbation of disease in 1 (2.1%) patient No effect of anti-copper treatment on pregnancy outcome	Of untreated WD group: 62 children: 4 (6.4%) were born at full term with body weight under 2500 g 3 (4.8%) were born with congenital heart disease 3 (4.8%) were born premature **Treated group**: 21 deliveries 1 (4.8%) with body weight under 2500 g 3 (14.3%) born premature 1 (4.8%) with cerebral palsy
Walshe (1986) [18]	7 patients and 11 pregnancies Retrospective analysis	All treated with trientine (1200–2000 mg) Mean duration of treatment: 5 years (1 newly diagnosed and treatment started at beginning of pregnancy)	**Pregnancy outcome**: Successful: 8/11 (72.7%) Premature: 1/11 (9.0%) Abortion: 1/11 (9.0%) Therapeutic termination: 1/11 (9.0%) **Deliveries**: 6 spontaneous; 3 Cesarean sections **Maternal outcome**: No effect of pregnancy on mothers’ health; even 1 extremely ill patient at time of conception who started anti-copper treatment was well at delivery.	**Birth defects**: 1/9 (11.1%) child with isochromosome X (premature birth)
Walshe (1977) [19]	10 patients and 15 pregnancies Retrospective analysis	Neurological WD: 3 Hepatic: 3 Presymptomatic: 4 9 patients treated with DPA 500–2000 mg (at the time of conception) Mean duration: 10 years (range 2.5–19 years) During pregnancy: 6 continued DPA without breaks, 7 interrupted treatment until after 12 weeks Untreated presymptomatic: 1 1 uncertain	**Pregnancy outcome**: Successful: 12/15 (80.0%) Premature membranes rupture: 2/15 (13.3%) Therapeutic terminations at 12 weeks: 1/15 (6.6%) **Deliveries**: 10 natural; 3 Cesarean sections; 1 forceps delivery; 1 surgical induction **Maternal outcome**: No aggravation of WD symptoms	**Birth defects**: None
Marecek and Graf (1976) [7]	8 patients and 12 pregnancies Retrospective analysis	7 women treated with DPA for 1–7 years DPA was discontinued in 5 during pregnancy, 2 before pregnancy	**Pregnancy outcome**: Successful 12 pregnancies (100.0%) pregnancies terminated with Cesarean sections due to fetal problems **Deliveries**: 10 natural; 2 Cesarean sections **Maternal outcome**: WD severity did not change in 7 cases; exacerbation of hepatic disease in 1 case (previously with advanced liver disease)	**Birth defects**: None
Scheinberg and Sternlieb (1975) [14]	18 patients and 29 pregnancies Retrospective analysis	Asymptomatic: 4 Hepatic WD: 3 Neurological WD: 8 Other symptoms: 3 Multiple spontaneous abortions: 2 Secondary amenorrhea: 1 All treated with DPA 900 –1500 mg (from 1 to 16 years) Stopped DPA (first trimester): 2	**Pregnancy outcome**: Successful: 29 (100.0%) No detailed data on 2 patients with multiple spontaneous abortions **Deliveries**: Type not provided **Maternal outcome**: No exacerbation of 1 case of pre-eclampsia	**Birth defects**: None

DPA = d-penicillamine; WD = Wilson’s disease.

**Table 2 biomedicines-10-02072-t002:** Summary of case and series reports of pregnant patients with Wilson’s disease according to the time of diagnosis, anti-copper treatment, pregnancy and delivery complications, maternal and newborn outcomes, and birth defects.

Authors	Patient Characteristics	Outcome: Pregnancy and Maternal	Outcome: Newborn
Saito et al. (2019) [31]	A 24-year-old woman with WD since aged 21 years (hepatic form with severe liver failure, qualified for liver transplant despite DPA).	**Pregnancy outcome**: Alive (gestosis during pregnancy) and Cesarean section at week 30 due to pregnancy complications **Delivery**: Cesarean section due to medical indications **Maternal outcome**: Gestosis from 25 week of pregnancy, liver failure, obstetric DIC, nephrotoxicity Improvement after 40 days of delivery	**Birth defect**: Low body weight (1173 g), mechanical ventilation, further discharged home without complications
Wan et al. (2018) [32]	A 26-year-old woman treated for 15 years with DPA, replaced with zinc 3 months before pregnancy.	**Pregnancy outcome**: Successful **Delivery**: Cesarean section on request **Maternal outcome**: Due to low platelet (56,000/mm^3^), general anesthesia was performed during the Cesarean section. No WD aggravation.	**Birth defect**: None
Durairaj et al. (2018) [33]	A 32-year-old woman with WD diagnosis established during pregnancy (hepatic and neurological symptoms). Zinc started from 28 weeks of pregnancy.	**Pregnancy outcome**: Recurrent hematemesis; premature newborn (32 week) **Delivery**: Natural **Maternal outcome**: Transient worsening of liver functions tests	**Birth defects**: Low weight of newborn 1860 g; transient mechanical ventilation
Kurdi et al. (2017) [30]	A 33-year-old woman with hepatic WD (diagnosed 9 months before) with liver cirrhosis and portal hypertension.	**Pregnancy outcome**: Abortion **Maternal outcome**: No WD aggravation	Abortion
Avcioglu et al. (2015) [26]	A 24-year-old woman with WD for 7 years (liver cirrhosis) treated with DPA 900 mg/day and zinc 100 mg, stopped 3 days before end of pregnancy.	**Pregnancy outcome**: Successful **Delivery**: Cesarean section due to medical indication **Maternal outcome**: Preeclampsia; HELLP syndrome (qualified for liver transplantation after delivery)	**Birth defects**: None
Lee et al. (2015) [28]	A 33-year-old woman with WD for 20 years, treated for 18 years (stopped 2 years before pregnancy). Introduced zinc oxide (dose not provided) again during pregnancy.	**Pregnancy outcome**: Successful **Delivery**: Natural **Maternal outcome**: Severity of WD did not change during pregnancy	**Birth defects**: None
Malik et al. (2013) [39]	Patient 1: A 30-year-old woman with neurological WD, treated with zinc sulfate 150 mg/day for 8 years. Patient 2: A 33-year-old woman with neurological WD, treated with zinc sulfate for 4 years. Patient 3: A 21-year-old woman with psychiatric WD for 3 years, treated with zinc sulfate (2 pregnancies).	**Pregnancy outcome**: Successful **Delivery**: 3 natural; 1 Caesarean section (Patient 2, due to preeclampsia) **Maternal outcome**: Good outcome for Patients 1 and 3 No WD aggravation	**Birth defects**: None
Rich et al. (2012) [42]	A 35-year-old woman with psychiatric symptoms (pregnancies at age 32 and 35) treated with DPA (dose not provided) and switched to zinc in first pregnancy; during the second (at 34 weeks switched for DPA until delivery at 36 week) and additionally on lithium monotherapy (mood stabilizers).	**Pregnancy outcome**: Successful **Deliveries**: 2 natural **Maternal outcome**: Aggravation of psychiatric symptoms after both deliveries	**Birth defects**: None
Masciullo et al. (2011) [41]	A 36-year-old woman with neurologic WD, treated with zinc acetate (150 mg/day) for 2 years; zinc dose reduced to 100 mg/day during pregnancy.	**Pregnancy outcome**: Successful **Delivery**: Natural **Maternal outcome**: No WD aggravation	**Birth defects**: None
Czlonkowska et al. (2010) [35]	Patient 1: A 33-year-old woman (previously healthy) without prior WD diagnosis, admitted to hospital due to placental abruption. Patient 2: A 30-year-old woman who experienced neurological symptoms (speech disorders) in the third trimester, two days before delivery.	**Patient 1: Pregnancy outcome**: Successful **Delivery**: Cesarean section **Maternal outcome**: 12 h after surgery experienced HELP syndrome, hemorrhagic stroke, hemiparesis, poststroke-epilepsy persists **Patient 2: Pregnancy outcome**: Successful **Delivery**: Cesarean section **Maternal outcome**: 3 h after surgery experienced HELLP syndrome, neurological symptoms exacerbated	**Birth defects**: None in either patient
Theodoridis et al. (2009) [36]	A 28-year-old clinically asymptomatic woman, treated with DPA 1000 mg/day for 4 years. Decreased dose to 500 mg from 6 weeks of pregnancy.	**Pregnancy outcome**: Antepartum hemorrhage with placenta abruption **Delivery**: Cesarean section due to medical indication (antepartum hemorrhage) **Maternal outcome**: No WD aggravation	**Birth defects**: None
Hanukoglu et al. (2008) [43]	A DPA-treated woman (age and history unknown) with 2 pregnancies.	**Pregnancy outcome**: Successful **Deliveries**: Natural **Maternal outcome**: No WD aggravation	**Birth defects**: Hypothyroidism (duration of 7 months in first child and 4 years in second child); low body weight 2400 g
Pinter et al. (2004) [49]	A 35-year-old woman treated with DPA 1000 mg/day; dose reduced to 500 mg/day since 20 weeks of pregnancy.	**Pregnancy outcome**: Oligohydroamnios **Delivery**: Natural **Maternal outcome**: No WD aggravation	**Birth defects**: Diffuse cutis laxa, micrognathia, agenesis of corpus callosum, 4-limb contractures
Furman et al. (2001) [38]	A 33-year-old woman diagnosed at aged 17 years, treated with DPA 500 mg/day (5 pregnancies at age 23, 24, 25, 28 and 33).	**Pregnancy outcome**: Successful **Deliveries**: Natural **Maternal outcome**: No WD aggravation during first 4 pregnancies; portal hypertension and esophageal varices during the last	**Birth defects**: None
Mustafa et al. (1998) [51]	A 28-year-old woman with a history of 9 spontaneous abortions. During the tenth pregnancy, WD diagnosis was established; initiated DPA up to 800 mg/day.	**Pregnancy outcome**: 9 spontaneous abortions before treatment; 5 successful **Deliveries**: 1 Cesarean section (medical indication); 4 natural **Maternal outcome**: Improved neurological symptoms	**Birth defects**: 1 newborn with low weight (2250 g); 4 cases none
Borghella et al. (1997) [47]	A 30-year-old woman with neurological WD, diagnosed 12 years before. Treated with DPA 750 mg/day with dose decreased to 500 mg/day at 24 weeks of pregnancy. Received diazepam for anxiety.	**Pregnancy outcome**: Successful **Delivery**: Natural **Maternal outcome**: No WD aggravation	**Birth defects**: None
Devesa et al. (1995) [45]	A 22-year-old woman with hepatic WD, treated with trientine 1000 mg/day for 9 years.	**Pregnancy outcome**: Successful **Delivery**: Cesarean section due to medical conditions (42 week of gestation) **Maternal outcome**: No WD aggravation	**Birth defects**: None
Nunns et al. (1995) [46]	A woman with neurological WD, diagnosed at 14 years old and treated with DPA (1500 mg/day). Dose reduced to 500 mg/day during first pregnancy (at 17 years old). Dose reduced to 500 mg/day during second pregnancy (at 20 years old) from 20 weeks of pregnancy.	**Pregnancy outcome**: Successful **Deliveries**: 1st natural; 2nd Cesarean section due to medical indication **Maternal outcome**: Aggravation of neurological symptoms during first pregnancy; neurological deterioration and portal hypertension (hepatic worsening) during second pregnancy. Symptoms recovered in both cases.	**Birth defects**: None
Hartard and Kunze (1994) [37]	A 32-year-old woman with neurological WD for 1 year, treated with DPA 1200 mg and zinc sulfate (800 mg), which continued during pregnancy.	**Pregnancy outcome**: Successful first delivery before WD diagnosis at age 17 without maternal and pregnancy complications **Delivery**: 2 natural **Maternal outcome**: No WD aggravation	**Birth defects**: None
Oga et al. (1993) [22]	A 23-year-old woman diagnosed with WD at 12 years old, treated with DPA for 2 years after diagnosis then untreated for 9 years.	**Pregnancy outcome**: Alive (gestosis during pregnancy) **Delivery**: Cesarean section **Maternal outcome**: Successful	**Birth defects**: Low birth weight (2380 g) and hepatomegaly (but further normal development)
Shimono et al. (1991) [20]	A 28-year-old woman, treated with DPA for 14 years. First two pregnancies on WD treatment without complications, then treatment stopped by patient at conception of third pregnancy.	**Pregnancy outcome**: Alive **Deliveries**: Spontaneous **Maternal outcome**: Fatal (acute liver failure occurred on day of delivery)	**Birth defects**: None
Chin (1991) [23]	Patient 1: Treated with zinc sulfate Patient 2: A 24-year-old on DPA 1000 mg/day	**Pregnancy outcome**: Alive (fetal distress in Patient 2) **Deliveries**: 1 spontaneous; 1 Cesarean section (Patient 2) **Maternal outcome**: Successful	**Birth defects**: Low birth weights (2700 g and 1560 g)
Van Leeuwen et al. (1991) [55]	A 27-year-old woman with neurological symptoms (7 spontaneous abortions before WD diagnosis when aged between 21 and 26 years), treated with zinc sulfate, next pregnancies at age 27 and 31 years.	**Pregnancy outcome**: 7 spontaneous abortions when untreated; 2 successful when treated **Delivery**: 2 natural **Maternal outcome**: No WD aggravation	**Birth defects**: None
Soong et al. (1991) [50]	A woman with hepatic WD, treated with DPA 1000 mg/day for 2 years.	**Pregnancy outcome**: Successful **Delivery**: Natural **Maternal outcome**: No WD aggravation	**Birth defects**: None
Dupont et al. (1990) [21]	A 25-year-old woman treated with DPA 750 mg/day for 4 years (dose not changed during pregnancy).	**Pregnancy outcome**: Alive **Deliveries**: Natural **Maternal outcome**: No WD aggravation	**Birth defects**: None
Lao et al. (1988) [52]	A 29-year-old woman with neurological WD; not treated for 7 years (apart from splenectomy) then received zinc treatment, which was stopped before the fourth pregnancy.	**Pregnancy outcome**: 2 spontaneous abortions without anti-copper treatment; 2 healthy children **Deliveries**: Natural **Maternal outcome**: No WD aggravation	**Birth defects**: None
Martinez-Frias et al. (1988) [54]	A 22-year-old woman treated with DPA	**Pregnancy outcome**: Successful **Delivery**: Natural **Maternal outcome**: No WD aggravation	**Birth defects**: Cleft lip and palate
Morimoto et al. (1986) [53]	A 22-year-old woman with WD for 5 years, treated with DPA 1200 mg. DPA was stopped before pregnancy and re-introduced at 500 mg/day in the 14th week of pregnancy.	**Pregnancy outcome**: Alive **Delivery**: Cesarean section **Maternal outcome**: Successful	**Birth defects**: None
Biller et al. (1985) [24]	A 31-year-old woman with neurological WD for 4 years, treated with DPA 1500 mg. DPA dose was reduced to 1000 mg in the first trimester.	**Pregnancy outcome**: Successful **Delivery**: Natural **Maternal outcome**: No WD aggravation	**Birth defects**: None
Linares et al. (1979) [44]	A 24-year-old woman with two spontaneous abortions before diagnosis (clinically symptomatic). Then treated with DPA 1500 mg/day, which was continued during pregnancy.	**Pregnancy outcome**: 2 spontaneous abortions; 1 successful **Delivery**: 1 natural **Maternal outcome**: No WD aggravation	**Birth defects**: Reversible cutis laxa
Fukuda et al. (1977) [25]	Patient 1: A 20-year-old woman with hepatic WD (liver cirrhosis), treated with DPA 1500 mg/day for 10 years without interruption. Patient 2: A 32-year-old woman with neurological WD for 7 years; DPA stopped 4 years before pregnancy then re-introduced (with BAL) 2–3 months before pregnancy.	**Pregnancy outcome**s: Successful **Deliveries**: Natural **Maternal outcome**s: No WD aggravation	**Birth defects**: None
Toaff et al. (1977) [29]	Patient with neurological WD, treated with DPA 750 mg since 14-years old. Two pregnancies at aged 21 and 23 years.	**Pregnancy outcome**: Successful **Deliveries**: Natural **Maternal outcome**: High blood pressure in the first pregnancy; no complications in the second. No WD aggravation.	**Birth defects**: None
Albukerk (1973) [34]	A 36-year-old woman with neurological WD, treated with DPA 1200 mg irregularly (4 pregnancies at aged 21, 27, 31 and 36 years).	**Pregnancy outcome**: 2 spontaneous abortions (aged 31 and 36 years); 2 successful **Deliveries**: 2 natural **Maternal**: No WD aggravation	**Birth defects**: None
Dreifuss and McKinney (1966) [52]	A 27-year-old woman with neurological WD, treated for 1 year with dimercaprol injections 10 days/each month (frequency reduced during pregnancy).	**Pregnancy outcome**: 2 successful **Deliveries**: 2 natural **Maternal outcome**: No WD aggravation	**Birth defects**: None
Sherwin et al. (1960) [40]	A 24-year-old woman with neurological WD, treated with DPA 900 mg/day (6 days/week) for 1 year; DPA stopped in last trimester.	**Pregnancy outcome**: Successful **Delivery**: Natural **Maternal outcome**: Improvement of neurological symptoms	**Birth defects**: None
Bihl (1959) [27]	A 22-year-old woman with neurological WD (diagnosed in 1955); treated initially with courses of calcium disodium versenate and BAL, which were discontinued due to adverse events.	**Pregnancy outcome**: 2 spontaneous abortions; 1 successful **Deliveries**: 1 natural **Maternal outcome**: heavy loss of blood in the second abortion; gestosis in the third pregnancy, hyperemesis with nephrotoxic signs	**Birth defects**: None

DPA = d-penicillamine; ZS = zinc salts; WD = Wilson’s disease; BAL = British anti-lewisite; DIC = disseminated intravascular coagulation. The data presented on pregnant WD women differed, especially in retrospective studies where findings were presented more as summaries, with tables describing analyzed patients in detail (in smaller studies). In case reports, data were often presented by gynecologists who did not fully describe WD, its treatment or WD-related maternal outcomes (e.g., neurological and hepatological symptoms). Moreover, it was difficult to distinguish the patients who stopped anti-copper treatment during pregnancy to perform additional analyses on how discontinuation may have impacted the maternal and pregnancy outcomes.

## Data Availability

Not applicable.

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
