# Peer review of "The Maternal and Fetal Outcomes of Pregnancy in Wilson’s Disease: A Systematic Literature Review and Meta-Analysis"

_biomedicines, 2022, doi:10.3390/biomedicines10092072_

Round 1
Reviewer 1 Report
The analysis presented by the authors is very interesting and can be of considerable help to doctors in managing and advising pregnant women with Wilson's disease. However, some minor revisions are needed and reported below.
1. Already in the abstract, the sum of the two percentages (78.3% and 21.6%) is not 100%. Please check and correct.
2. At the end of the introduction authors reported 54 retrospective studies and case/series reports (references 7-60). Differently, in the abstract, they reported 49 publications. Check and correct the mismatch.
3. In Table 1, in the first reference (Mussi et al 2021) fetal malformations are reported to be 6/18. Why 18? Are considered only treated pregnancies? This is not clear.
4 Throughout the manuscript the percentages are reported with non-uniform rounding. Sometimes a single decimal place is shown, other times none. Moreover, 0.5 is sometimes rounded up and sometimes down. Please check the text and tables and correct any percentages reported incorrect.
5. In Table 1, in Reuner and Dinger (2019) in “Patient characteristics” the numbers don't add up and/or are not clearly reported. Please correct.
6. Throughout Table 1, better manage the sub-lists, especially as regards treatments, otherwise the numbers reported are not immediately clear as they should be, and the treatment subgroups as well.
7. In Table 1, in Scheinberg and Sternlieb (1975) 29 pregnancies are reported, all successful. 2 patients with multiple spontaneous abortions are reported too. Are these extra pregnancies than those reported? If yes, please report it better. If not, the pregnancy numbers don't add up.
8. Table 1 is not very uniform. Clearly, the same types of data cannot be reported since they are retrospective studies, but try as much as possible to schematize it better and more uniformly.
9. Even if reported immediately after the reference text, please put the wording of the figures in the correct position in the text (both figure 1 and figure 2).
10. Authors reported “aggravation of neurological symptoms in 2% of pregnancies (18/822) and of hepatic symptoms in 4.6% (38/822).” However, it is not specified whether there is a correlation of worsening with the presence/absence of treatment. Evaluate this aspect.
11. In the meta-analysis in Figure 2, please also reported the total number of pregnancies and the number of abortions.
Author Response
Thank You very much for your letter on our manuscript entitled: “The maternal and fetal outcomes of pregnancy in Wilson’s disease: a systematic literature review and meta-analysis”. Wee carefully analyzed the reviewer’s comments, and we implemented some changes according to their suggestions and discussed others. Below we shortly present major reviewer’s comments, our response and changes we made in our manuscript.
RESPONSE TO THE REVIEWERS
Reviewer 1
Dear Reviewer, Thank you very much for your review and comments. Below we respond to your comments on our article.
Reviewer #1
COMMENTS TO THE AUTHOR:
Reviewer 1
The analysis presented by the authors is very interesting and can be of considerable help to doctors in managing and advising pregnant women with Wilson's disease. However, some minor revisions are needed and reported below.
- Already in the abstract, the sum of the two percentages (78.3% and 21.6%) is not 100%. Please check and correct.
Answer:
Thank You for the comment, we corrected it in abstract and results.
- At the end of the introduction authors reported 54 retrospective studies and case/series reports (references 7-60). Differently, in the abstract, they reported 49 publications. Check and correct the mismatch.
Answer:
Thank You for the comment, we corrected it as well as list of references through the manuscript. The correct number is 13 studies and 36 case and series reports (49 publications). References numbers 7 till 55.
- In Table 1, in the first reference (Mussi et al 2021) fetal malformations are reported to be 6/18. Why 18? Are considered only treated pregnancies? This is not clear.
Answer:
Thank You for the comment. From 26 pregnancies there were 8 abortions, hence there were 18 pregnancies and 6 fetal malformations. In tables abortions are provided as pregnancy outcome and birth defects are described in the column newborn outcome.
4 Throughout the manuscript the percentages are reported with non-uniform rounding. Sometimes a single decimal place is shown, other times none. Moreover, 0.5 is sometimes rounded up and sometimes down. Please check the text and tables and correct any percentages reported incorrect.
Answer:
Thank You for the comment, we checked again the article and corrected it, in the abstract we rounded it up, however in Tables with direct results we provided the data with decimal place.
- In Table 1, in Reuner and Dinger (2019) in "Patient characteristics" the numbers don't add up and/or are not clearly reported. Please correct.
Answer:
Thank You for the comment, it could be misleading, We corrected it:
There were 22 patients. Fourteen (30% dose reduction) treated with DPA, but 1 discontinued treatment during pregnancy due to the death of mother and fetus). Four treated with trientine, one with zinc and 3 non treated not diagnosed with WD
- Throughout Table 1, better manage the sub-lists, especially as regards treatments, otherwise the numbers reported are not immediately clear as they should be, and the treatment subgroups as well.
Answer:
We clarified this data as possible, most of the data presented is very important for physicians involved in WD treatment and these data were described often non homogenously.
- In Table 1, in Scheinberg and Sternlieb (1975) 29 pregnancies are reported, all successful. 2 patients with multiple spontaneous abortions are reported too. Are these extra pregnancies than those reported? If yes, please report it better. If not, the pregnancy numbers don't add up.
Answer:
Thank You for the comment, Scheinberg and Sternlieb described in details 18 women and 29 pregnancies, showing successful deliveries, however in their paper two patients 17 and 18 were mentioned only in Table 1, with no detailed data; so we provided such information in Table 1, as well as in study limitations.
We added in study limitations:
“Also in one of the oldest studies presenting the successful outcome of 29 pregnancies in 18 WD patients, in two patients the data about multiple spontaneous abortions were provided (without details) [14] what caused that we could not add them to analysis”.
- Table 1 is not very uniform. Clearly, the same types of data cannot be reported since they are retrospective studies, but try as much as possible to schematize it better and more uniformly.
Answer:
We clarified this data as possible, most of the data presented is very important for physicians involved in WD treatment and these data were described often non homogenously.
- Even if reported immediately after the reference text, please put the wording of the figures in the correct position in the text (both figure 1 and figure 2).
Answer:
We corrected the positions of Tables and Figures as requested.
- Authors reported "aggravation of neurological symptoms in 2% of pregnancies (18/822) and of hepatic symptoms in 4.6% (38/822)." However, it is not specified whether there is a correlation of worsening with the presence/absence of treatment. Evaluate this aspect.
Answer:
Due to retrospective nature of the article, these data were not available to sum up. Most of the studies presented maternal outcome/complications separately presenting just number of patients and outcome. So we couldn’t provide more detailed data in this interesting topic. We added this in study limitation.
The added text is:
Due to limited data, an analysis of fetal and maternal outcomes was not possible among patients who stopped treatment during pregnancy versus those who continued treatment.
- In the meta-analysis in Figure 2, please also reported the total number of pregnancies and the number of abortions.
Answer:
Thank You for the comment – we corrected it. Improved Figure 2 was provided.
Reviewer 2 Report
This study is the first systematic review and meta-analysis of the literature on pregnancy-related outcomes in female patients with WD over a period of more than 60 years. The findings reaffirm the importance of continuing anti-copper therapy in pregnant WD patients. Although the article suffers from a large heterogeneity of study subjects, a long time span, and some missing data, the overall statistics on pregnancy in WD patients provide directional guidance for patients and physicians.
There is indeed the difficulty of reviewing the data and information of a large number of articles one by one to do such a literature review analysis. One thing I am more concerned about is that the methods section of the article does not explicitly list the exclusion criteria. The study dropped from 3529 records to 480 after excluding duplicates, but the culling of the remaining 431 articles was only summarized as a review of relevance; were these articles lacking statistically usable data? Or were the exclusions made for other reasons? The authors should complete this section as well as Figure 1.
Author Response
RESPONSE TO THE REVIEWERS
Dear Reviewer 2, Thank you very much for your review and comments. Below we respond to your comments on our article.
Answer:
We corrected Figure 1, providing the more clear and detailed flow diagram, with detailed reasons of article exclusions .
In methods section we additionally added the registration of our protocol as well as information about data checking:
Firstly, title and abstracts were screened according to study criteria independently by all authors (reviewers) – duplicate records were removed. Incomplete reports, reviews, editorial, commentaries, conference proceedings, discussions as well as overlapped data were excluded after extensive assessment and revisions.
Then, full texts were obtained and screened with the same mode.
“The study protocol was registered and got INPALSY registration number: INPALSY202280003 (doi: 10.37766/inpalsy2022.8.00003)”
We hope you will find it interesting enough to publish it in Biomedicines
With best regards,
Tomasz Litwin, MD, PhD tomlit@medprakt.pl
Agnieszka Antos MD, agantos@ipin.edu.pl
Prof. Anna Czlonkowska, MD, PhD czlonkow@ipin.edu.pl
Prof. Iwona Kurkowska-Jastrzębska MD, PhD ikurkowska@ipin.edu.pl
Prof. Adam Przybylkowski MD, PhD aprzybylkowski@interia.pl
Jan Bembenek MD, PhD jbembenek@o2.pl
Marta Skowrońska MD, PhD mskowronska@ipin.edu.pl
Lukasz Smolinski MD, PhD smolinski@ipin.edu.pl